# Movie Box Office Prediction Based on Multi-Model Ensembles

**Yuan Ni [1], Feixing Dong [2],\*, Meng Zou [2] and Weiping Li [3]**

[1] School of Economics and Management, Beijing Information Science and Technology University, Beijing 100192, China; niyuan@bistu.edu.cn

[2] Computer School, Beijing Information Science and Technology University, Beijing 100192, China; zoumeng@bistu.edu.cn

[3] School of Software & Microelectronics, Peking University, Beijing 102600, China; wpli@ss.pku.edu.en

\* Correspondence: feixingdong@bistu.edu.cn

**Abstract:** This paper is based on the box office data of films released in China in the past, which was collected from ENDATA on 30 November 2021, providing 5683 pieces of movie data, and enabling the selection of the top 2000 pieces of movie data to be used as the box office prediction dataset. In this paper, some types of Chinese micro-data are used, and a Baidu search of the index data of movie names 30 days before and after the release date, coronavirus disease 2019 (COVID-19) data in China, and other characteristics are introduced, and the stacking algorithm is optimized by adopting a two-layer model architecture. The first layer base learners adopt Extreme Gradient Boosting (XGBoost), the Light Gradient Boosting Machine (LightGBM), Categorical Boosting (CatBoost), the Gradient Boosting Decision Tree (GBDT), random forest (RF), and support vector regression (SVR), and the second layer meta-learner adopts a multiple linear regression model, to establish a box office prediction model with a prediction error, Mean Absolute Percentage Error (MAPE), of 14.49%. In addition, in order to study the impact of the COVID-19 epidemic on the movie box office, based on the data of 187 movies released from January 2020 to November 2021, and combined with a number of data features introduced earlier, this paper uses LightGBM to establish a model. By checking the importance of model features, it is found that the situation of the COVID-19 epidemic at the time of movie release had a certain related impact on the movie box office.

**Keywords:** stacking predictor; box office predictor; impact of COVID-19 epidemic

## 1. Introduction

From the 1950s when artificial intelligence started, it has been used for theorem proving. In the 1970s, expert systems appeared in the field of artificial intelligence, which combined reasoning with professional knowledge, making expert systems successful in medical, chemical, and other fields. The combination of artificial intelligence and the new terms "big data" and "cloud computing" have now enabled voice recognition and driverless driving technologies become reality. With the development of and improvement in artificial intelligence theory and technology, it has gradually been applied to various human production and living activities.

With the improvement in people's living standards and film production level, more and more people enrich their lives by watching movies. For the success of a movie, its box office is an important evaluation index, and how to reasonably and accurately predict the box office through data has become the focus of researchers. An effective box office prediction model can provide business decision support and guidance for film production and distribution companies, which is of great significance for the sustainable development of the film industry.

Machine learning is a research hotspot in the field of artificial intelligence, and its theories and methods have been widely applied to solve complex problems in engineering applications and scientific fields [1,2]. At present, there are few pieces of research on box

office prediction in China, and most of the existing methods use multiple linear regression. The prediction method of multiple linear regression is lagging behind other methods, and the prediction error of a single model is relatively high. At present, the prediction performance of a multi-model ensemble algorithm is relatively strong, and a multi-model ensemble can help improve the generalization ability of the model and reduce the prediction error of the model.

The contribution of this paper is mainly reflected in the following three aspects: (1) The amount of data used in the current research on the prediction of China's box office is relatively small, and the amount of data used in our study is relatively larger. (2) The majority of the existing research on box office influencing factors focuses on film characteristics. To study box office prediction, this paper combines micro-data, the Baidu search index, and epidemic data. (3) After experiments, the improved stacking algorithm adopted in this paper has a better prediction effect compared with multiple single-model algorithms and multiple model ensemble algorithms.

## 2. Related Research Work

Until now, academic research on movie box office prediction has mainly focused on the research of the movie box office prediction model and the influencing factors of the movie box office. In the early stages of the research on the movie box office prediction model, researchers mostly used the linear regression algorithm. Later, with the development of artificial intelligence, researchers gradually used various machine learning algorithms and neural networks to predict the movie box office. The research on the factors affecting the box office can be divided into two categories: static factors and dynamic factors. Static factors are the factors that have been determined before the film is released, such as the film name, director, actors, sequel, duration, production company, distribution company, schedule, film type, film language, film region, etc. Dynamic factors, such as a Weibo review, film review, the Baidu search index, film reputation, etc., are the factors that will affect the box office when the film is about to be released and has been released.

In terms of box office prediction model research, Wang Jinhui and Yan Siyu [3] established a box office prediction model based on the Litman model [4] and the Byeng-Hee Chang model [5] by using multiple linear regression analysis. Wenbin Zhang and Steven Skiena [6] used the Lydia news analysis system to predict the movie box office. After their analysis, it was found that the model using only news data achieved a similar performance to the model using IMDB data, and better performance could be achieved by combining them; a regression model is more suitable for low box office movies, while the KNN model is more suitable for high box office movies. Han Zhongming et al. [7] proposed a GBRT model under the same data conditions, and the prediction effect was better than those of the random forest, decision tree, and nonlinear regression models. Li Jian Ping and Wang Shimin [8] analyzed the influencing factors of the Chinese box office through grey correlation degree analysis, selected important factors, and made their prediction using a BP neural network, and this achieved a good prediction effect. Zheng Jian and Zhou Shangbo [9] proposed a film box office prediction model based on a feedback neural network, which had a good effect in predicting box office classification. Ru Yunian et al. [10] established a pre-screening box office prediction model based on deep learning, including a cross network and a dense network, which was superior to SVM and the random forest algorithm in the case of the same feature.

For the study of static factors, a relatively complete quantitative system of influencing factors has been formed at present. For example, Wang Zheng and Xu Min [11] take the traditional OLS method and set the different critical values of the film; using the logit regression model, and considering the production quality, the types of films, and a variety of factors such as national income level, their research showed that the stars, director, score, prices, sequel, and schedule will have a positive impact on the box office. Based on the capital theory of sociologist Bourdieu, Zhao Xinxing and Gao Fuan [12] made a comparative analysis of three regression results for the total box office, first week box office,

and follow-up box office. The study showed that the director, leading actor, whether it was a sequel, and the IP adaptation with variables of economic capital and cultural capital played a positive role, while real events played a negative role. Among the variables of social capital, word-of-mouth and policy factors play a significant role in promoting the box office of theme films. Based on a two-dimensional analysis of static and dynamic perspective, Chen Haoshu [13] divided the influencing factors into film release return, film release risk control, and film investment decision, and applied a structural equation and system dynamics model, respectively, to empirically analyze the impact of influencing factors on the world film box office in the short and long term. The analysis results showed that the film investment decision has the greatest impact on the world film box office. Awards and distribution companies have a significant positive correlation with the world film box office, while piracy has a significant negative correlation. In the long term, the impact of distribution risk control is relatively volatile. Bae and Kim [14] studied the impact of movie titles on box office success, and the research results showed that informative movie titles had a positive impact on the box office income of films with insufficient publicity, but with the increase in promotional activities before release, such an impact would decrease. The dynamic factors affecting the box office mainly include the number of movie reviews, what the movie reviews say, the movie score, Weibo comments, Baidu search index, and so on.

With the development of the Internet, Weibo and Douban have gradually developed and become new platforms for people to discuss and spread movie information. More and more people pay attention to public opinion information on the Internet, which makes the impact of online word-of-mouth information on the box office more and more important. Many Chinese scholars have studied the impact of Douban reviews and Weibo reviews on the box office. For example, Hua Rui et al. [15] used the C–D production function to measure the impact of various influencing factors on the box office. The study shows that word-of-mouth information will significantly promote box office growth, but with the box office growth, the positive impact of word-of-mouth information will also decline, and the promotional effect of modern propaganda methods on the box office is greater than that of traditional propaganda methods. Pei and Jiang Yinyan [16] established a mathematical model of box office appeal using a questionnaire survey. Empirical results show that data related to film brands, such as word-of-mouth score have a greater impact on the box office, while the story adaptation, 3D technology, production company, and media publicity have less impact on the box office. Jiang Zhaojun et al. [17] studied the relationship between online word-of-mouth information and the box office of animated films. Through the establishment of a multiple regression model, the test shows that the number of online word-of-mouth users, that is, the number of Douban-rated users, has obvious positive effects on the box office of animated films, and after the first week of release, it has a greater positive effect on the follow-up box office of animated films. Online word-of-mouth potency can only improve the box office of imported animated films, and the positive effect on the subsequent box office of imported animated films is stronger after the first week of release. Du et al. [18] used microblog data to predict the box office, which is based on two sets of features: counting-based features and content-based features. For the former, they describe the characteristics from the user's point of view, pay attention to the different values of different Weibo reviews, and eliminate the impact of junk users. Based on the features of content, they put forward a new semantic classification method for the box office, which makes features more relevant to the box office.

Basuroy et al. [19] investigated how movie reviews affect box office performance, and how stars and budgets adjust this impact. The research shows that in the first week of the film release, the impact of bad reviews on the box office is greater than that of good reviews, and the impact of bad reviews on films will weaken with the passage of time. Popular stars and high budgets will boost the box office of films with bad reviews more than good reviews, but the effect on films with good reviews more than bad reviews is minimal, which provides insights for film companies on how to strategically manage film

reviews to improve the box office. Gaenssle et al. [20] conducted a study on the influencing factors of the box office success of Russian international films in 2012–2016, which provides a supplement to the research of the Russian film market. The box office success factors are divided into distribution-related factors, brand and star effects, and evaluation sources. Gaenssle et al. added regional variables, such as seasonality, the time span between the world and Russia, the attendance of international stars at the premiere of Russian films, and the adaptation of titles in line with Russian culture. The study found that the reviews of international film critics and the adaptation of film titles were most important. Lee and Choeh [21] studied the impact of comment validity on the box office, and analyzed and verified the impact of comment quantity, comment grade, comment length, and comment usefulness on the Korean box office. Hyunmi Baek [22] studied the impact of the word-of-mouth information of different types of social media on the box office at different stages of film release. In the early stages, Twitter has a greater impact on the box office revenue, while Yahoo has a greater impact on the box office in the later stages. The impact of blogs and YouTube varies little between the early and later stages. Lei Sun et al. [23] investigated the impact of movie consumers' willingness on movie marketing and the box office, and the willingness was measured by the number of people who wanted to see a film. This impact is tested by the step-by-step method and generalized least square method based on random effects, and it is found that consumer willingness and box office revenue have an inverted U-shaped distribution with event marketing intensity. From the second week before the release to the first week after the release, marketing activities have a significant positive impact on the box office. Ram Sharda and Dursun Delen [24] transformed the box office prediction problem into a classification problem, not predicting the point estimate of box office revenue, but classifying according to the box office revenue of a film. According to the characteristics of the film's score, competition, star, film type, technical effect, whether it is a sequel or not, and the number of screens, the neural network is used for modeling, and good results are achieved. Biramane et al. [25] collected the data of movie features and social interaction from various social platforms (such as IMDb, YouTube, and Wikipedia), and established a prediction model by establishing the relationship between classic features, social media features, and movie box office success. The results showed that the prediction model established by combining classic and social factors has high accuracy.

At present, the research foundation of the influencing factors of film consumption and the box office prediction model is solid, and the evaluation framework of influencing factors, including the chief producer team, film characteristics, and word-of-mouth reviews, has been formed. Research methods are gradually extended to machine learning, neural networks, data-driven approaches in the context of big data, etc. In this paper, based on fully considering the complex and comprehensive influencing factors in the digital era, market information, gross national product, and other information will be introduced, and a multi-model ensemble method will be adopted for modeling in the research method to improve the effectiveness of the evaluation model.

## 3. Establishment of Prediction Model

### 3.1. Data Collection

The data used for modeling in this paper come from ENDATA, the Easy Professional Superior (EPS) data platform, the Chinese National Health and Health Commission, and the Baidu Index.

Movie information was downloaded and crawled from the movie box office information on the website of ENDATA, including the title of the movie, the number of people who want to watch the movie on MaoYan, the number of people who want to watch the movie on Taobao Film, the number of people who want to watch the movie on Douban, the movie's score on MaoYan, the movie's score on Taobao Film, the movie's score on Douban, duration, actors, screenwriters, synopsis, release date, the annual number of movie screenings, the total number of movie screenings, the annual audience (10,000), the total audience (10,000), box office in the first week (10,000), the average ticket price, the

number in the audience for an average screening, main genre, complete genre, director, production company, distribution company, movie format, country, investment scale, integrated marketing company, new media marketing company, producer, executive producer, publisher, movie attribute, service fee (10,000), and total box office (10,000).

Chinese micro-data on the EPS data platform was downloaded, including the consumption level of all residents (yuan), the consumption level of rural residents (yuan), the consumption level of urban residents (yuan), the consumption level index of all residents (1978 = 100), the consumption level of rural residents (1978 = 100), Gross Domestic Product (GDP) (100 million yuan), the proportion of the tertiary industry (GDP = 100), per capita disposable income of residents (yuan), cinema lines, screens in cinema lines (block), number of Internet users (10,000), Internet broadband access ports (10,000), Internet international export bandwidth (Mbps), proportion of administrative villages with Internet broadband services (%), Internet penetration rate (%), Internet broadband access users (10,000).

Since 1 January 2020, the National Health Commission (NHC) has started to report on the pneumonia cases of the novel coronavirus infection in China. In order to study whether the epidemic has had an impact on the box office, we searched the daily epidemic news on the NHC website since the outbreak. Currently, search engines have become an important way for consumers to know about movies. We used the Baidu search index every day from 30 days before the release to 30 days after the release according to the movie name for box office prediction.

### 3.2. Data Feature Processing and Analysis

By deleting duplicate data and missing box office data in ENDATA, 5683 film data were left after processing, and the top 2000 films were used for box office prediction modeling after integrating the missing situations of other features of films. By checking the distribution of the film box office, illustrated in Figure 1, it can be found that compared with the normal distribution, the distribution of the film box office is more consistent with the unbounded Johnson distribution. Thus, the value of the box office must be converted before regression.

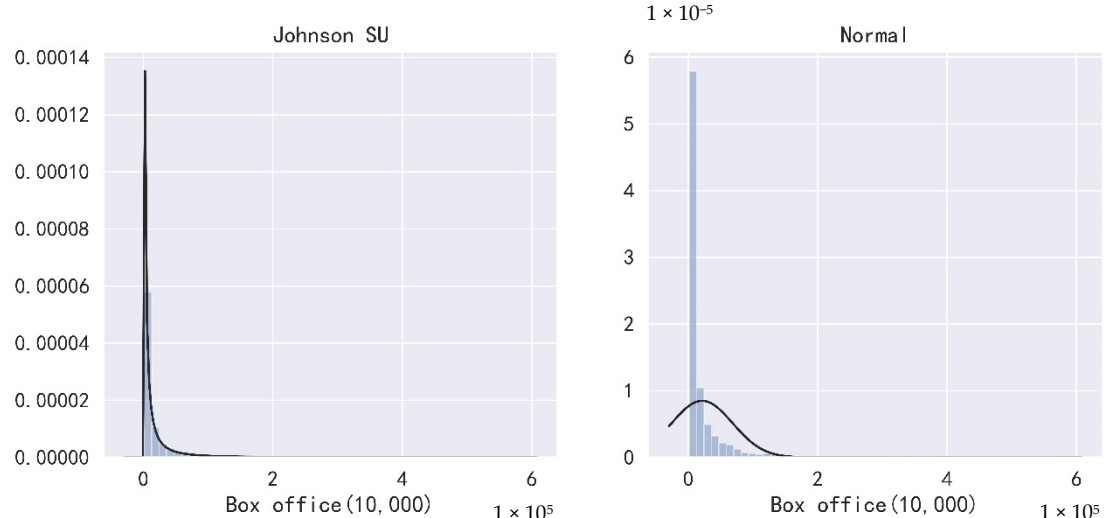

**Figure 1.** Distribution of total box office (10,000).

### 3.2.1. Chinese Micro-Data Processing

Micro-data of 16 kinds of national statistics were obtained from the EPS website, and each data were annual data. Polynomial regression was used to fit each kind of data according to the existing data, and the microscopic data of each film were calculated according to the polynomial regression model after fitting. As there are many kinds of microscopic data, the data of screens in cinema lines (block) is taken as an example to

illustrate how to use polynomial regression for prediction. The specific prediction process is as follows:

Step 1: The screens in cinema lines (block) are the data from 2006 to 2020, and the year is converted into the last date of that year, and then the date is converted into a number (1 January 1900 is the number 1, 2 January 1900 is the number 2, and so on). Examples of conversions are shown in Table 1.

**Table 1.** Examples of data of screens (blocks) in cinemas.

| Year | Date | Converted Date to Number | Screens in Cinema Lines (Block) |
|------|------|--------------------------|---------------------------------|
| 2015 | 31 December 2015 | 42,369 | 31,600 |
| 2017 | 31 December 2017 | 43,100 | 50,776 |
| 2019 | 31 December 2019 | 43,830 | 69,787 |

Step 2: With the date number as the independent variable and the number of screens (blocks) in the cinema line as the dependent variable, python was used to establish a polynomial regression model. The highest power of the polynomial was set as 5, and the $R^2$ of the model was 0.9972. The fitting curve drawn according to the real value and the predicted value is illustrated in Figure 2.

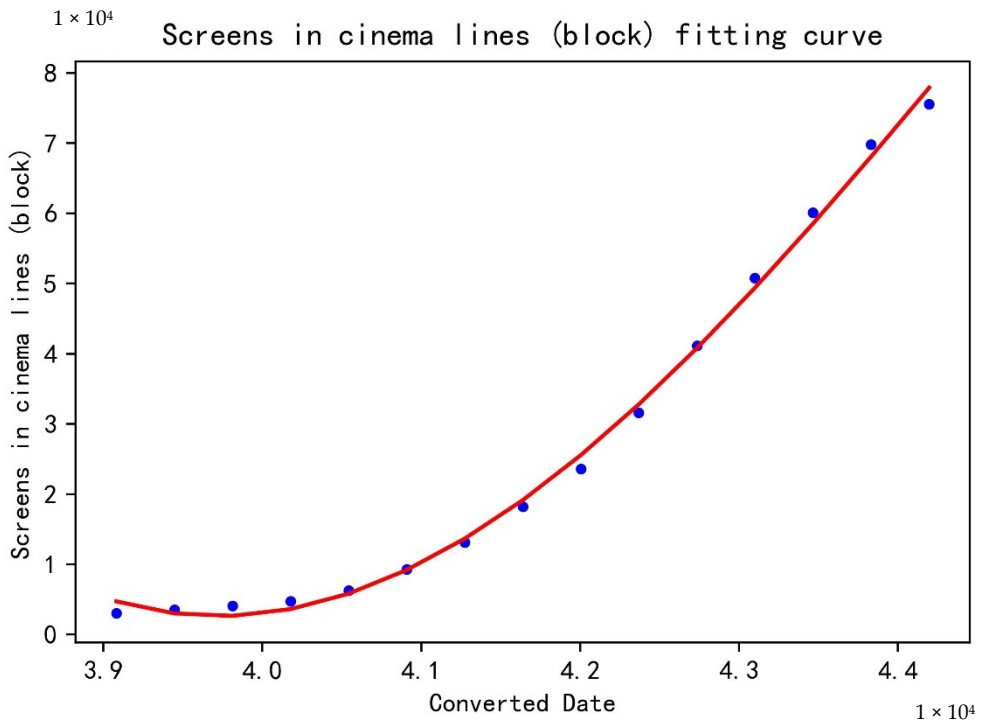

**Figure 2.** Screens in cinema lines (block) fitting curve.

Step3: With the release date of each movie as the independent variable, the polynomial regression model obtained in Step2 is used to predict the in-cinema screen (block) of the movie when the movie is released. Part of the predicted data is shown in Table 2.

Through the above steps, the polynomial regression model is used to train and predict each kind of microscopic data, and the predicted value of microscopic data when each film is released is taken as the data feature of box office prediction.

**Table 2.** Example of Screens (Block) Predict Data in Cinema Line.

| Movie Name | Release Date | Convert the Release Date to Number | Predicted Value of Screen (Block) in Cinema Line |
|---|---|---|---|
| The Battle at Lake Changjin | 30 September 2021 | 44,469 | 85,346.67536984 |
| Wolf Warriors 2 | 27 July 2017 | 42,943 | 45,625.00855624 |
| Hi, Mom | 12 February 2021 | 44,239 | 79,071.3646575 |
| Ne Zha | 26 July 2019 | 43,672 | 63,895.13502012 |
| The Wandering Earth | 5 February 2019 | 43,501 | 59,454.47257928 |

3.2.2. Epidemic Data Processing

On the website of the National Health Commission, we can search the daily news about the epidemic since the outbreak. We used regular expressions to extract for each day information on newly confirmed cases, existing confirmed cases, new asymptomatic infection, accumulated cured discharged cases, cumulative death cases, reported cumulative confirmed cases, existing suspected cases, cumulative tracking of close contacts, and close contacts under medical observation; some examples of the epidemic data are shown in Table 3.

**Table 3.** Examples of some epidemic data.

| Date | Newly Confirmed Cases | Existing Confirmed Cases | New Asymptomatic Infection | Accumulated Cured Discharged Cases | Cumulative Death Cases | Reported Cumulative Confirmed Cases | Existing Suspected Cases | Cumulative Tracking of Close Contacts | Close Contacts under Medical Observation |
|---|---|---|---|---|---|---|---|---|---|
| 19 August 2021 | 46 | 1866 | 30 | 88,044 | 4636 | 94,546 | 2 | 1,154,405 | 41,706 |
| 18 August 2021 | 28 | 1887 | 17 | 87,977 | 4636 | 94,500 | 1 | 1,153,308 | 43,156 |
| 17 August 2021 | 42 | 1928 | 17 | 87,908 | 4636 | 94,472 | 1 | 1,151,965 | 44,471 |
| 16 August 2021 | 51 | 1938 | 20 | 87,856 | 4636 | 94,430 | 1 | 1,150,082 | 45,325 |
| 15 August 2021 | 53 | 1922 | 29 | 87,821 | 4636 | 94,379 | 1 | 1,146,611 | 44,300 |

We added epidemic data features to the data of each movie based on the epidemic data and the release date of the film every day during the epidemic period, including newly confirmed cases on the day of release, existing confirmed cases on the day of release, asymptomatic infected persons on the day of release, cured and discharged cases on the day of release, dead cases on the day of release, reported confirmed cases on the day of release, tracked close contacts on the day of release, close contacts who are still under medical observation on the day of release, newly confirmed cases one week before release, newly confirmed cases 30 days before release, newly confirmed cases one week after release, and newly confirmed cases 30 days after release.

3.2.3. Baidu Search Index Data Processing

According to the movie name, we added Baidu search index features for each movie data, including the average search index 30 days before release, average search index 7 days before release, search index on release day, average search index 7 days after release, average search index 30 days after release, average search index 30 days before release on PC, average search index 7 days before release on PC, search index on release day on PC, average search index 7 days after release on PC, average search index 30 days after release on PC, average search index 30 days before release on intelligent terminal, average search index 7 days before release on intelligent terminal, search index on release day on intelligent terminal, average search index 7 days after release on intelligent terminal, average search index 30 days after release on intelligent terminal, and Baidu search index for each of the 30 days before and after the release. The examples of the Baidu search index data of some movies after sorting are shown in Table 4.

Table 4. Some Baidu search index data examples of movies.

| Movie Name | Release Date | Average Search Index 30 Days before Release | Average Search Index 7 Days before Release | Search Index on Release Day | Average Search Index 7 Days after Release | Average Search Index 30 Days after Release |
|---|---|---|---|---|---|---|
| The Battle at Lake Changjin | 30 September 2021 | 30,008.7 | 72,595.43 | 502,677 | 1,059,134 | 451,876.7 |
| Wolf Warriors 2 | 27 July 2017 | 24,158.23 | 45,348.71 | 256,913 | 1,262,703 | 920,833.4 |
| Hi, Mom | 12 February 2021 | 37,625.4 | 104,796.7 | 395,390 | 748,152.4 | 309,665.4 |
| Ne Zha | 26 July 2019 | 16,633.3 | 63,845 | 283,248 | 539,412.1 | 331,797.2 |
| The Wandering Earth | 5 February 2019 | 31,611.17 | 53,317 | 471,650 | 1,099,486 | 516,540.5 |

3.2.4. Director, Actor, and Other Characteristic Data Processing

In order to measure the box office impact of each movie's director, actor, writer, producer, executive producer, producer, production company, manufacturing company, distribution company, integrated marketing company, and new media marketing company, the following characteristics were derived by collating all movie data.

We set the first three actors of each film as the leading actors, and sorted out all the film data according to the characteristics of the film itself, such as the leading actor, screenwriter, director, publisher, executive producer, producer, production company, production company, distribution company, integrated marketing company, and new media marketing company. We added the number of these subjects in a film, the total number of films historically involved, the total box office in the past, and the average box office in the past for each film.

For each film's director, screenwriter, and each leading actor, we added the features of the total number of the audience, on MaoYan, who wanted to watch the movies that these people had worked on in the past, the highest number of the audience, on MaoYan, who wanted to watch the movies that these people had worked on in the past, the average number of the audience, on MaoYan, who wanted to watch the movies that these people had worked on in the past, the total number of the audience, on Taobao Film, who wanted to watch the movies that these people had worked on in the past, the highest number of the audience, on Taobao Film, who wanted to watch the movies that these people had worked on in the past, the average number of the audience, on Taobao Film, who wanted to watch the movies that these people had worked on in the past, the total number of the audience, on Douban, who wanted to watch the movies that these people had worked on in the past, the highest number of the audience, on Douban, who wanted to watch the movies that these people had worked on in the past, the average number of the audience, on Douban, who wanted to watch the movies that these people had worked on in the past, the total score of the movies, on MaoYan, that these people had worked on in the past, the highest score of the movies, on MaoYan, that these people had worked on in the past, the average score of the movies, on MaoYan, that these people had worked on in the past, the total score of the movies, on Taobao Film, that these people had worked on in the past, the highest score of the movies, on Taobao Film, that these people had worked on in the past, the average score of the movies, on Taobao Film, that these people had worked on in the past, the total score of the movies, on Douban, that these people had worked on in the past, the highest score of the movies, on Douban, that these people had worked on in the past, the average score of the movies, on Douban, that these people had worked on in the past.

We added the first director (i.e., ranking the first director) for each movie data, the first screenwriter, the first producer, the first production company, the first manufacturing company, the first distribution company, and calculated the number of movies that they had worked on in the past, the total box office of movies that they had worked on in the past, the average box office of movies that they had worked on in the past, the total box office of movies that they had worked on in the last year, the number of movies that they had worked on in the last year, the average box office of the movies that they had worked on in the last year, the total box office of the movies that they had worked on in the last 3 years,

the number of movies that they had worked on in the last 3 years, the average box office of the movies that they had worked on in the last 3 years, the total box office of the movies that they had worked on in the last 5 years, the number of movies that they had worked on in the last 5 years, the average box office of the movies that they had worked on in the last 5 years.

Based on all the movie data, we calculated the involvements and number of times of all actors, screenwriters, directors, publishers, executive producers, production companies, manufacturing companies, distribution companies in movies, to participate in the movie number sorting before the one percent threshold, to confirm the list of model actors, model scriptwriter, model director, model producer, model supervisor, model producer, model production company, model production company, and model distribution company, and, according to the actual work of each movie, to determine the number of model actors, model scriptwriters, model directors, model publishers, model executive producers, model producers, model production companies, model manufacturing companies, and model distribution companies, and add these data to the movie box office data features.

### 3.2.5. Feature Processing of Release Date Data

According to the release date of movies, we added the release year, release date number, and release day of the week to each movie's data. The release date feature is a string. The first four characters are obtained by slicing the string to obtain the release year feature of the movie. Release date numbers are obtained by converting release dates to numbers in Excel. Use the strptime function of the datetime library to convert the release date to something similar to %Y/%m/%d, then call weekday and +1 to obtain the day of the week. Figures 3 and 4 show the distribution of the release year and the day of the week of the top 2000 movies at the box office.

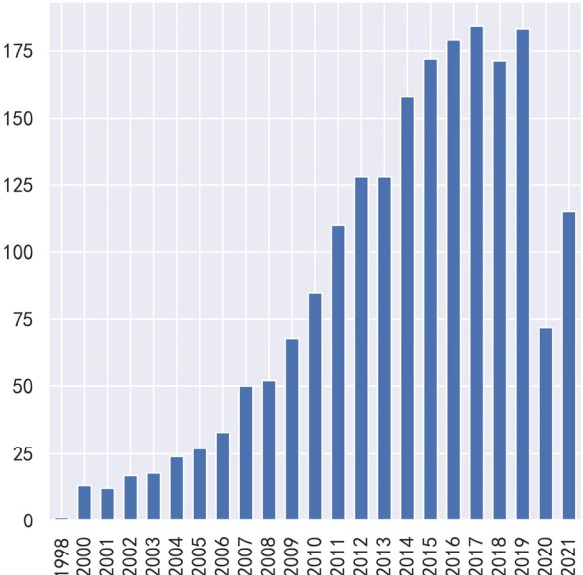

**Figure 3.** Release year distribution pie chart.

In addition, schedule features were included for each movie, including New Year's Day, Spring Festival, Valentine's Day, Women's Day, Qingming Festival, Labour Day, Children's Day, Dragon Boat Festival, summer, Qixi Festival, Mid-Autumn Festival, National Day, and New Year's Day. The specific schedule time range is illustrated in Table 5. The schedule of each movie was manually marked in Excel according to the release date of the movie and the time range of each time period.

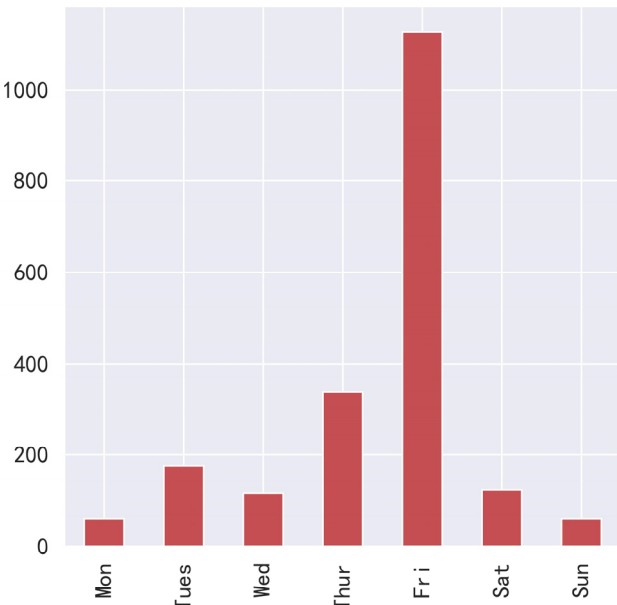

**Figure 4.** Which day of the week is the release date?

**Table 5.** Movie schedule time range.

| Movie Schedule | Time Range |
|---|---|
| New year's day | From the end of December of each year to 3 January of the following year |
| Spring Festival | Spring Festival holiday every year |
| Valentine's Day | Every year from 13 February to 14 February |
| Women's day | Every year from 7 March to 8 March |
| Qingming Festival | Every year on Tomb-Sweeping Day. |
| Labour Day | Every year from the end of April to 5 May |
| Children's Day | Every year from the end of May to 1 June |
| Dragon Boat Festival | Dragon Boat Festival holiday every year |
| Summer Vacation | Every year from the beginning of June to the end of August |
| Chinese valentine's day | Every year Chinese Valentine's Day |
| Mid-autumn Festival | Mid-Autumn Festival holiday every year |
| National Day | Every year from the end of September to 7 October |
| New Year movie season | Every year from the end of November to the end of December |
| Ordinary | Outside the time range of other schedules |

### 3.2.6. Processing of Other Data Features

We added the data features of the title length, the number of words in the plot brief, and the number of movie types for each movie, divided the complete type of each movie into type_1, type_2, and type_3, and manually added the features of whether the movie is a series.

Before entering the model, LabelEncoder was used to encode the features of all types of data (such as time schedule, investment scale, main type, film format, country, film attribute, etc.), and the mode was used to fill in the missing values of type features. Secondly, all numerical features were standardized to make the distribution of numerical features conform to the standard normal distribution. Then, the missing values were filled with 0, which is equivalent to the mean filling of the missing values.

### 3.3. Model Establishment

The model used in this paper is optimized on the basis of the stacking algorithm. The stacking algorithm is the technique of training a model to combine other models (base learner). Firstly, train some different base learners. Secondly, the second layer meta-learner is trained with the output of the base learners as the input of the meta-learner. The stacking

algorithm can take advantage of the power of a range of well-performing base learners for classification or regression tasks, which, better than any base learner, is a good way to decrease the variance model.

The first layer of stacking prediction models is to decrease the variance by training different types of models on the same dataset, each of which is different and produces different results. According to the results generated by each base learner in the first layer as the input features of the second layer meta-learner, the final regression results are obtained after training.

In the first layer, a simple average method is used to select the results of the k-fold cross-validation of each base learner. If the accuracy of the base learner generated by each training is high, the proportion of correct datasets is high, and the final prediction result is close to the true value. Conversely, if the number of incorrect datasets accounts for a large proportion, the noise of the input set at the second layer is higher; this can decrease the accuracy of the final result and a simple average may cause the precision loss. Thus, to enhance the stacking strategy, this research uses every single fold model's $R^2$ to simulate "accuracy". The weight of the base learner of the model with a poor prediction effect was decreased by implementing the precision difference of stacking, and the movie box office prediction model was established based on the enhanced stacking strategy.

To take into account the size of data samples and the feature missing rate, the research data of this model are the data of the top 2000 films at the box office. The modeling process is as follows:

Step 1: Firstly, the processed data are divided into a training set and test set in a ratio of 9:1.

Step 2: Establish the model with LightGBM, and obtain the importance of each feature in the LightGBM model. Delete the features with importance less than three, and retain the Chinese national micro-data, epidemic data, and Baidu search index features.

Step 3: Build the first layer-based learner model, and build the XGBoost, LightGBM, Cat-Boost, GBDT, random forest and SVR models based on deleted features. During the construction of the first layer-based learner model, the training set obtained by Step1 is divided into five folds for training by adopting the method of five-fold cross-validation, and the goodness of fit of each training is obtained.

Step 4: Build the second layer meta-learner model. The second layer meta-learner selects a multiple linear regression model and connects the prediction results of the training set of each base learner in the first layer as the training set of the meta-learner for the training of the meta-learner. The goodness of fit of each fold prediction model of each base learner in the first layer is used to simulate the "precision", and the weighted average of the prediction results of the test set is used as the test set of the meta-learner.

The algorithm flow is shown in Figure 5.

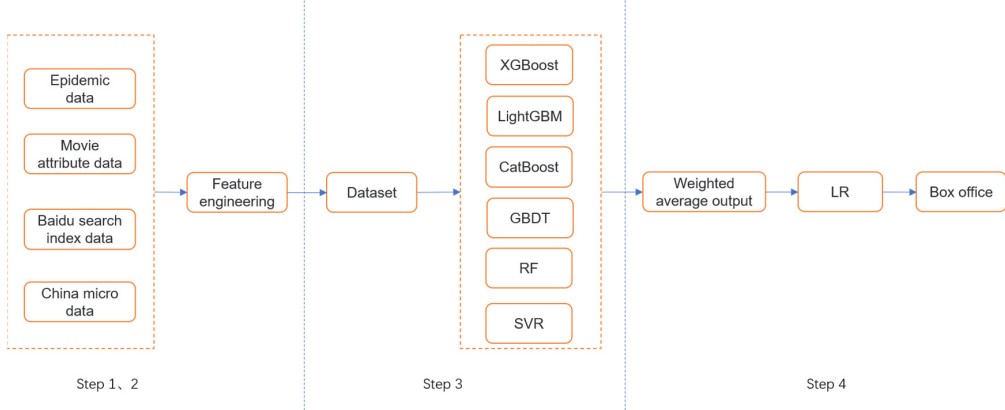

**Figure 5.** Stacking algorithm architecture diagram.

*3.4. Experimental Results and Analysis*

3.4.1. Evaluation Indicators

In this paper, $R^2$, Mean Absolute Percentage Error (MAPE), Mean Absolute Error (MAE) and Mean Square Error (MSE), Root Mean Square Error (RMSE) are used to evaluate the prediction effect of the model.

(1)　For $R^2$, the range is [0, 1]. If the result is 0, it indicates that the model fitting effect is poor. If the result is 1, the model fits perfectly. The larger $R^2$ value is, the better the model fitting effect is.

$$R^2 = \frac{\sum\limits_{i=1}^{n} (\hat{y}_i - \overline{y})^2}{\sum\limits_{i=1}^{n} (y_i - \overline{y})^2} \tag{1}$$

(2)　MAPE represents the relative error between the predicted value and the real value. The smaller the MAPE value is, the higher the prediction accuracy is and the smaller the error is.

$$MAPE = \frac{100\%}{n} \sum\limits_{i=1}^{n} \left| \frac{\hat{y}_i - y_i}{y_i} \right| \tag{2}$$

(3)　MAE represents the average absolute error between the predicted value and the real value. The smaller the MAE value is, the better the model effect is.

$$MAE = \frac{1}{n} \sum\limits_{i=1}^{n} |(y_i - \hat{y}_i)| \tag{3}$$

(4)　MSE is the mean of the sum of squares of errors between the predicted value and the true value. The smaller the MSE value is, the better the model effect is.

$$MSE = \frac{1}{n} \sum\limits_{i=1}^{n} (y_i - \hat{y}_i)^2 \tag{4}$$

(5)　RMSE is the square root of MSE, and the smaller the RMSE value, the better the model effect.

$$RMSE = \sqrt{\frac{1}{n} \sum\limits_{i=1}^{n} (y_i - \hat{y}_i)^2} \tag{5}$$

In the above formula, $\hat{y}_i$ represents the $i$th predicted value of the model, $y_i$ represents the $i$th true value, $\overline{y}$ represents the average of all box offices, and $n$ represents the number of data.

3.4.2. Model Comparison

In order to evaluate the prediction effect of the model, four models, such as ordinary stacking, XGBoost, LightGBM, and CatBoost, were selected for comparison experiments, and the average absolute percentage error of the model established in this paper was compared, as shown in Table 6.

**Table 6.** Model prediction error.

| Model | $R^2$ | MAPE | MAE | MSE | RMSE |
|---|---|---|---|---|---|
| XGBoost | 0.8578 | 17.96% | 5010.3662 | 366,253,785.4177 | 19,137.7581 |
| LightGBM | 0.8993 | 16.19% | 4204.9473 | 259,371,807.7103 | 16,105.0243 |
| CatBoost | 0.8709 | 18.19% | 5063.2543 | 332,490,044.7863 | 18,234.3096 |
| GBDT | 0.8503 | 16.41% | 4972.4056 | 385,492,351.2144 | 19,633.9591 |
| RF | 0.7812 | 21.27% | 6650.2541 | 563,602,015.1131 | 23,740.3036 |
| SVR | 0.6531 | 69.82% | 12,680.4396 | 893,514,540.6887 | 29,891.7136 |

**Table 6.** *Cont.*

| Model | $R^2$ | MAPE | MAE | MSE | RMSE |
|---|---|---|---|---|---|
| Voting | 0.8420 | 20.14% | 5829.9820 | 406,872,334.5275 | 20,171.0767 |
| Averaging | 0.8476 | 19.13% | 5524.3003 | 392,511,674.4714 | 19,811.9074 |
| Ordinary Stacking | 0.8726 | 14.52% | 4168.6142 | 328,196,923.9439 | 18,116.2061 |
| Enhanced Stacking | 0.8746 | 14.49% | 4143.7905 | 323,085,161.6079 | 17,974.5699 |

Common model ensemble algorithms include voting, averaging, bagging, boosting, and stacking. In order to evaluate the prediction effect of the model, we select several single models and multiple model ensemble algorithms, including XGBoost, LightGBM, CatBoost, GBDT, RF, SVR, voting, averaging, ordinary stacking, and enhanced stacking, for comparison experiments. By observing the prediction effect of the model under different hyperparameters, the optimal hyperparameters of each model are selected.

(1) XGBoost is based on the gradient boosting algorithm. Compared with other implementations of the gradient boosting algorithm, the execution speed of XGBoost is very fast. The hyperparameters of the XGBoost algorithm used in this paper are as follows: n_estimators = 200, learning_rate = 0.08, subsample = 0.8, max_depth = 5, and colsample_bytree = 0.9.

(2) LightGBM, like XGBoost, is an efficient implementation of GBDT, with a faster training speed, lower memory consumption, and other advantages. The hyperparameters of the LightGBM algorithm used in this paper are as follows: boosting_type = 'gbdt', learning_rate = 0.1, min_child_samples = 20, n_estimators = 200, and num_leaves = 31.

(3) CatBoost, which is also an improved implementation under the framework of the GBDT algorithm, is mainly characterized by efficient and reasonable processing of category features. The hyperparameters of CatBoost algorithm used in this paper are as follows: iterations = 200 and loss_function = 'RMSE'.

(4) GBDT, which is a representative algorithm of the boosting algorithm, is an optimal model obtained by iterative training with a weak learner (decision tree) and has the advantages of good training effect and difficulty in over-fitting. The hyperparameters of the GBDT algorithm used in this paper are as follows: max_depth = 5, n_estimators = 200, subsample = 0.8, and learning_rate = 0.1.

(5) RF is the representative algorithm of the bagging algorithm. Sampling is carried out in the way of putting back. Sub-models are established with the sampled samples and trained on the sub-models. The hyperparameters of the RF algorithm used in this paper are as follows: n_estimators = 200 and bootstrap = True.

(6) SVR is an application of the support vector machine (SVM) to regression problems. The basic idea is to fit the optimal hyperplane. The hyperparameter of the SVR algorithm used in this paper is kernel = 'RBF'.

(7) Voting can help improve the generalization ability of models and reduce the error rate of models. For the regression model, voting's final prediction is an average of the results predicted by multiple other regression models. The single models used in the voting model in this article are XGBoost, LightGBM, CatBoost, GBDT, RF, and SVR, whose hyperparameters are shown above.

(8) Averaging means the results of multiple single models are weighted and averaged. We determined the weights based on the $R^2$ values of each model. The single models used by the averaging model in this paper are XGBoost, LightGBM, CatBoost, GBDT, RF, and SVR, whose hyperparameters are described above.

(9) For the ordinary stacking model, the first layer used in this paper includes XGBoost, LightGBM, CatBoost, GBDT, RF, and SVR, and the second layer of the meta-learner is LR. The hyperparameters of the base learner are as shown above. The meta-learner uses default parameters.

(10) For the enhanced stacking model, the first layer used in this paper includes XGBoost, LightGBM, CatBoost, GBDT, RF, and SVR, and the second layer of the meta-learner is

LR. The hyperparameters of the base learner are as shown above. The meta-learner uses default parameters.

The experimental environment for the model is Jupyter Notebook, python 3.8.8, scikit-learn 0.24.1, matplotlib 3.5.1, pandas 1.2.1, numpy 1.21.5, xgboost 1.5.2, lightgbm 3.3.2, and catboost 1.0.4. The computer platform used in the experiment is the Windows 11 operating system, AMD Ryzen 5 5600H with Radeon Graphics, and 16 GB memory. The comparison results of the evaluation indicators of each model are shown in Table 6.

### 3.5. Example of Partial Box Office Prediction

Table 7 shows the box office predictions for some of the movies in the test set.

**Table 7.** Some movie's box office prediction results.

| Film Title | Year | Total Box Office (10,000 yuan) | Predict Box Office | MAPE Error |
|---|---|---|---|---|
| 47 Meters Down: Uncaged | 2020 | 4760.74 | 5009.039599 | 5.22% |
| D-War | 2008 | 3100.0 | 2656.583856 | 14.30% |
| Doraemon: Shin Nobita no Nippon tanjou | 2016 | 10,364.31 | 9381.766633 | 9.48% |
| Magic Mirror 2 | 2018 | 2063.43 | 2563.248472 | 24.22% |
| London Has Fallen | 2016 | 33,981.65 | 36,959.946839 | 8.76% |
| The Great Wall | 2016 | 117,456.91 | 136,822.815602 | 16.49% |
| Detective Chinatown | 2015 | 82,441.94 | 94,299.604909 | 14.38% |
| Lie Detector | 2021 | 2788.05 | 3027.54925 | 8.59% |

In conclusion, the film box office prediction error constructed by multi-model integration based on enhanced post-stacking proposed in this paper is smaller, the influencing factors are more comprehensive, and the experimental data set is larger, which is of certain significance for the development of the film industry.

### 3.6. Correlation between Epidemic Factors and Box Office

In order to study the correlation between the epidemic factors and the box office, we screened the data of 187 movies released from January 2020 to November 2021 from the data used above. The same data processing process was adopted, and then the LightGBM algorithm was used for modeling. The $R^2$ of the model is 0.8686, The MAPE of the model is 18.94%. The feature_importances_ function in the LightGBM module was used to obtain the feature importance of the model, and the top 50 most important features were obtained by sorting the feature importance. The feature importance bar chart is illustrated in Figure 6. The important features of LightGBM and the features' description are shown in Table 8. It can be seen that the newly confirmed cases one week after release, the existing suspected cases, and the newly confirmed cases 30 days after release all had a great impact on the box office.

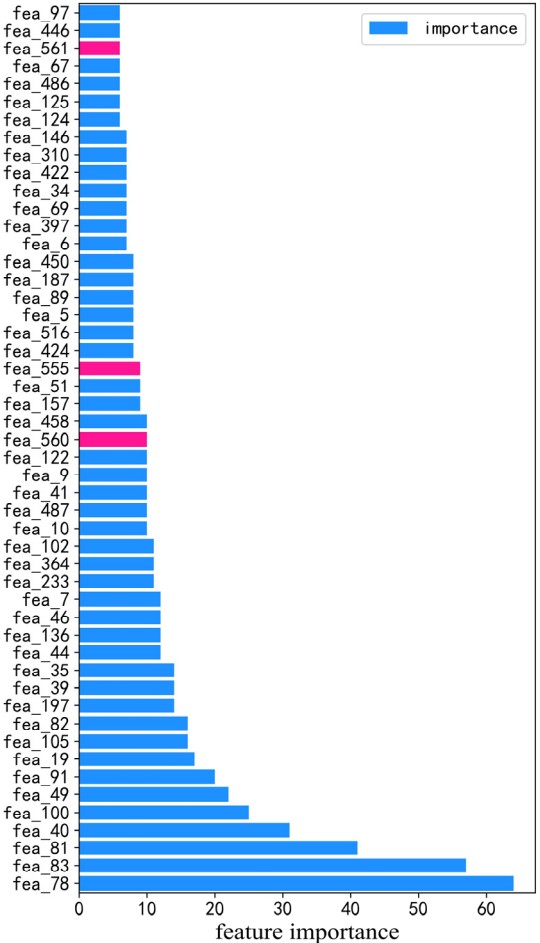

**Figure 6.** Importance of features of LightGBM model.

**Table 8.** Important features of LightGBM and features' description.

| Feature | Description | Feature | Description |
|---|---|---|---|
| fea_97 | Starring2 all-time total box office | fea_122 | The highest number of audience, on MaoYan, who wanted to watch the movies that leading actor3 had worked on in the past |
| fea_446 | Baidu search index of PC on the 20th day after movie releasing | fea_9 | The movie's score on Douban |
| fea_561 | Newly confirmed cases 30 days after the movie releasing | fea_41 | Production company average box office in the past |
| fea_67 | Release date number | fea_487 | Baidu search index of intelligent terminal on movie release day |
| fea_486 | Baidu search index of intelligent terminal on the 1st day before movie releasing | fea_10 | Duration |
| fea_125 | The average number of audience, on MaoYan, who wanted to watch the movies that leading actors had worked on in the past | fea_102 | Leading actor3 average box office in the past |
| fea_124 | The total number of audience, on MaoYan, who wanted to watch the movies that leading actors had worked on in the past | fea_364 | Baidu search index of all end on the 1st day before movie releasing |
| fea_146 | The total number of audience, on Douban, who wanted to watch the movies that leading actors had worked on in the past | fea_233 | The all-time total box office of integrated marketing company1 |
| fea_310 | The average box office of the distribution company1 over the past 5 years | fea_7 | The movie's score on MaoYan |
| fea_422 | Baidu search index of PC on the 4th day before movie releasing | fea_46 | Manufacturing company average box office in the past |

**Table 8.** *Cont.*

| Feature | Description | Feature | Description |
|---|---|---|---|
| fea_34 | Producer total box office in the past | fea_136 | The highest number of audience, on MaoYan, who wanted to watch the movies that leading actors had worked on in the past |
| fea_69 | Main genre | fea_44 | Manufacturing company total box office in the past |
| fea_397 | Baidu search index of PC on the 29th day before movie releasing | fea_35 | The total number of movies that producers worked on in the past |
| fea_6 | The number of people who want to watch the movie on Douban | fea_39 | Production company total box office in the past |
| fea_450 | Baidu search index of PC on the 24th day after movie releasing | fea_197 | The total number of audience, on Douban, who wanted to watch the movies that screenwriters had worked on in the past |
| fea_187 | The total number of audience, on Douban, who wanted to watch the movies that director had worked on in the past | fea_82 | Average ticket price |
| fea_89 | The number of genre | fea_105 | Leading actor's average box office in the past |
| fea_5 | The number of people who want to watch the movie on Taobao film | fea_19 | Director total box office in the past |
| fea_516 | Baidu search index of intelligent terminal on the 29th day after movie releasing | fea_91 | Genre_2 |
| fea_424 | Baidu search index of PC on the 2nd day before movie releasing | fea_49 | Distribution company total box office in the past |
| fea_555 | Existing suspected cases | fea_100 | Leading actor3 all-time total box office in the past |
| fea_51 | Distribution company average box office in the past | fea_40 | Production company total movie number in the past |
| fea_157 | The total score of movies, on MaoYan, that leading actors had worked on in the past | fea_81 | Box office in the first week (10,000) |
| fea_458 | Baidu search index of PC on the 29th day before movie releasing | fea_83 | The average number of audience in all screenings |
| fea_560 | Newly confirmed cases one week after the movie releasing | fea_78 | Total number of movie screenings |

## 4. Summary

In this paper, market information, epidemic information, national micro-data and other information are introduced to study the influencing factors on the box office. In terms of market information, data such as the number and average box office of films participating in the past of film production companies, manufacturing companies, distribution companies, integrated marketing companies, and new media marketing companies are collated and calculated, and the Baidu search index data of film titles are introduced. In terms of epidemic information, a variety of data including newly confirmed cases on the day of screening, existing confirmed cases, newly asymptomatic infected persons, cumulative cured and discharged cases, cumulative deaths, cumulative reported confirmed cases, and existing suspected cases were introduced, as well as their derivative data. In terms of national micro-data, the polynomial regression equation is established by taking the number of screens, the number of cinemas, GDP, Internet penetration rate, and other micro-data over the years as dependent variables and time as the independent variable. According to the release date of the movie, we predict the number of screens and theaters when the movie is released, as a prediction of the data characteristics of the research box office.

In this paper, different from the existing single-model evaluation, the enhanced stacking algorithm is used to integrate multiple models to build a movie box office prediction model. The movie box office prediction model is trained and tested on the dataset of the top 2000 films used in this paper, and the final MAPE is 14.49%. It is superior to XGBoost, LightGBM, and other models. The experimental results show that the enhanced stacking model used in this paper has a certain application value, but there is still room for further research and improvement in algorithm improvement. The data volume of box office prediction used in this paper is not large enough, and it can be further expanded in the future to establish a more effective box office prediction model. Through the box office data during the epidemic period, LightGBM was used to model, and the characteristic

importance of the model showed that the epidemic had a certain correlation with the box office.

**Author Contributions:** Conceptualization, Y.N.; data curation, Y.N. and F.D.; funding acquisition, Y.N.; investigation, F.D., M.Z. and W.L.; methodology, Y.N. and F.D.; project administration, Y.N.; resources, F.D.; software, F.D.; supervision, Y.N.; validation, F.D.; visualization, F.D.; writing—original draft, F.D.; writing—review and editing, F.D. All authors have read and agreed to the published version of the manuscript.

**Funding:** This work is funded by the National Key R&D Program of China (2021YFF0900200).

**Institutional Review Board Statement:** Not applicable.

**Informed Consent Statement:** Not applicable.

**Data Availability Statement:** The data used in this study are available from the corresponding author upon reasonable request.

**Acknowledgments:** This work is supported by [high tech research and development center of the Ministry of Science and Technology of China]. Thanks to every member of the team for their contributions.

**Conflicts of Interest:** The authors declare no conflict of interest.

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
