# Peer review of "Movie Box Office Prediction Based on Multi-Model Ensembles"

_information, doi:10.3390/info13060299_

Round 1

Reviewer 1 Report

  • The introduction is very short
  • A summary at the end of “ Related research work” should be added at the end of the section
  • Need to write more about XGBoost, LightGBM, CatBoost, GBDT, Random Forest, 317 SVR, Ridge regression and Lasso regression algorithm.
  • Figure 6. The importance of features of LightGBM model is not clear, need to be in better presentation as well as clear and readable

Reviewer 2 Report

The paper analyses market information, epidemic information, and Chines micro data. The influencing factors of box office are investigated.  Data is well analyzed. Data distribution is analyzed. Predictive model is developed based on the stacking ensemble technic. 

However, a lot of incomprehensibilities are in the paper:

1) Feature processing: mean value is not the best solution for missing data imputation (l 287), because the distribution is changed. Please explain your choice

2) 3.2.5 - how actually data processing is implemented? Fig 3-4 are hard for reading

3) l 294-297 - please explain your sentence.

4) The predictive accuracy before LightGBM usage and after should be compared.

5) Introduction is too short. The main contribution is missed

6) Related works section should be improved. Why authors didn't discuss different ensembles and their efficiency? Bagging, boosting, stacking and cascading should be analyzed.

7) What is the purpose of Table 2? Why polynomial regression model is used?

8) Conclusion section should be extended too.

Reviewer 3 Report

The manuscript is concerned with movie box official prediction based on ensemble learning, which is interesting. It is relevant and within the scope of the journal. However, the manuscript, in its present form, contains several weaknesses. Adequate revisions to the following points should be undertaken in order to justify recommendation for publication.

1.      Grammatical issues in the manuscript should be corrected

2.      Figure showing the overall flowchart should be provided.

3.      Complete software and hardware specifications of the system used to perform the implementations must be reported.

4.      Evaluation criteria including MAE, RMSE, and R2 should be provided for better showing the performance of the proposed mode.

5.      Moreover, the manuscript could be substantially improved by relying and citing more on recent literatures about metaheuristic and soft computing techniques such as the followings. http://dx.doi.org/10.1007/s00477-022-02183-5, https://doi.org/10.1155/2020/2624547. Discussions about result comparison and/or incorporation of those concepts in your works are encouraged:

6.      Section 1 is too short. Sections 1 and 2 are recommended to be merged.

7.      Full names should be shown for all abbreviations in their first occurrence in texts. For example, SVR, LR, and MAPE in abstract, etc. Please go through the manuscript for the abbreviations.

8.      Fig. 1: A base-10 log scale for the x axis is recommended.

9.      Fig. 2 : The title for X axis should be replaced with “converted date”.

10.   Fig. 3 and 4: texts are overlapped. Please redraw for better visualization.

11.   Fig. 6: Notations of the features are recommended to replace the full name.

12.   Comparison between the current work and previous works, especially those used artificial intelligence, is highly recommended

13.   In the conclusion section, the limitations of this study, suggested improvements of this work and future directions should be highlighted.

14.   Some key parameters are not mentioned. The rationale on the choice of the particular set of parameters should be explained with more details.

Round 2

Reviewer 2 Report

The authors significantly improved the paper. Thank you and good luck.

Author Response

Thanks for your suggestions. We have checked and corrected the whole paper again.

Reviewer 3 Report

The comments have been carefully addressed. Accept as it is.

Author Response

(The authors gave the same response as above.)
